# Comparison of changes in corneal volume and corneal thickness after myopia correction between LASIK and SMILE

Anna Schuh[1], Carolin M. Kolb[2], Wolfgang J. Mayer[1], Efstathios Vounotrypidis[1], Thomas Kreutzer[1], Thomas Kohnen[2], Siegfried Priglinger[1], Mehdi Shajari[1,2]*, Daniel Kook[1,3]

1 Department of Ophthalmology, Ludwig-Maximilians-University, Munich, Germany, 2 Department of Ophthalmology, Goethe University, Frankfurt, Germany, 3 SMILE Eyes Eye Clinic Munich Airport, Munich, Germany

* gmsams2020@gmail.com

**Data Availability Statement:** The data underlying this study are available on figshare. DOI: 10.6084/m9.figshare.14443685.

## Abstract

Myopia is the most common refractive error. Surgical correction with laser is possible. LASIK and SMILE are the techniques currently most used. Aim of the study was to compare changes in corneal volume and thickness after the respective laser treatment. 104 eyes of 52 patients were matched based on refractive error into two equally sized groups, either treated with LASIK or SMILE. Measurements were obtained from the Scheimpflug camera (Pentacam) preoperatively and at 3 and 12 months postoperatively. 3 months postoperatively, the flapless SMILE procedure resulted in a significant overall greater loss of corneal volume (P < 0.01) and corneal thickness (P < 0.01) compared to LASIK. No significant difference was found when comparing the 3 to 12-months values in each group. Within the currently used ranges of refractive error correction, loss in central corneal thickness and corneal volume with SMILE is higher in comparison to LASIK. As greater loss in corneal volume and thickness might contribute to higher level of corneal instability maximum ranges of refractive error correction with SMILE should not supersede those set currently for LASIK until more long-term results on corneal ectasia are available for SMILE.

## Introduction

Spending more time outdoors can reduce the risk of myopia and slow the myopic shift in refractive error [1]. Due to the fact that children spend more and more time indoors, the number of patients with myopia is likely to increase in the future, and thus the number of patients needing refractive surgery to obtain spectacle independence. Different procedures are in use today to correct myopia or myopic astigmatism in the long term, for example, the commonly used photorefractive keratectomy, laser–assisted in situ keratomileusis (LASIK), and the most recent small-incision lenticule extraction (SMILE) procedure. The principles of LASIK and SMILE are similar. In both procedures, a certain part of the cornea is removed that leads to a flatter corneal curvature, lower corneal thickness, and consequently, lower corneal refractive power.

**Funding:** SMILE Eyes Eye Clinic provided support in the form of salary for author DK. The specific roles of this author are articulated in the 'author contributions' section. The funder had no role in study design, data collection and analysis, decision to publish, or preparation of the manuscript. No additional external funding was received for this study.

**Competing interests:** The authors have read the journal's policy and have the following competing interests: DK is employed by the commercial company SMILE Eyes Eye Clinic. There are no patents, products in development or marketed products associated with this research to declare. This does not alter our adherence to PLOS ONE policies on sharing data and materials.

LASIK is normally used to correct myopia up to -8.00 D and astigmatism up to 5.00 D. In this technique, a corneal flap is created with a microkeratome or femtosecond laser [2]. After lifting and folding the hinged flap, ablations on the central part of the exposed stromal bed are produced using an excimer laser. The flap is then repositioned and irrigated with physiologic salt solution [3]. Since it is claimed that flap-based LASIK surgery reduces corneal biomechanical properties [4], the perceived need for a less manipulating technique led to the development of flapless SMILE. In this procedure, a stromal lenticule is produced using a femtosecond laser. The lenticule is then extracted from the stroma through a small incision [5, 6].

To date, many studies have evaluated the visual outcomes of SMILE compared to LASIK. A recent study [7] found lenticule extraction technology to be superior in postoperative visual performance over a 2-year observation period compared to LASIK, whereas both techniques were equal regarding efficacy and safety. Another study by Khalifa et al. [8] also found SMILE to be safe and effective but the authors stated a trend toward under correction of astigmatic myopia. These findings were congruous to those from Kanellopoulos [9] who obtained better refractive results using topography-guided LASIK surgery. Regarding ultrastructural changes of the cornea after refractive interventions, Luft et al. [10] observed that reactive fibrosis was less marked after SMILE compared to LASIK in a human donor eye model, however after LASIK the stromal bed exhibited a smoother surface texture.

It should be noted that none of these studies compared alterations in corneal volume (CV) or corneal thickness (CT) between LASIK and SMILE. The purpose of our study was to investigate changes in these biomechanical parameters after correction of myopia or myopic astigmatism. From our dataset of 104 eyes, we compared 3- and 12-month outcomes after both procedures using Pentacam Scheimpflug measurements of the anterior eye segment.

## Methods

### Location, time and patient inclusion

The study protocol was reviewed and approved by the Institutional Review Board of the Department of Ophthalmology at the Ludwig-Maximilians-University Munich (Ethikkommission LMU München) and the tenets of the Declaration of Helsinki were followed throughout the study. All patients participated in the study voluntarily and were informed of their right to abandon it at any chosen time without having to provide a reason. Informed consent was obtained in written form.

Medical records of 26 patients (52 eyes) who underwent myopia correction with LASIK and 26 patients (52 eyes) treated with SMILE were evaluated for this study. The LASIK group was composed of 26 patients with preoperative refractive values that were comparable to those from the SMILE group. Surgeries were performed by one experienced surgeon (D.K.) based at the SMILE Eyes Eye Clinic Munich Airport, Germany. All eyes meeting the following criteria were included: age over 18 years, complete preoperative and postoperative data, stable refraction, no additional ocular diseases and corneal abnormalities such as keratoconus, no previous corneal surgery, and no active infections. Our aim was to fully correct preoperative manifest refraction in all eyes.

### Preoperative and postoperative assessment

The following preoperative measurements were all performed using the Scheimpflug camera Pentacam (OCULUS Optikgeräte GmbH, Wetzlar, Germany): sphere, cylinder, partial CV of regions with 3, 5, and 7 mm diameters from the apex and total CV for which a 10 mm diameter was assumed, as well as pachymetry at the apex (central corneal thickness, CCT) and at peripheric zones of 2, 4, 6, and 8 mm diameters. The examination process was standardized and performed in a windowless clinical assessment room. The patients were instructed to blink

repeatedly, then to focus and not to blink or move their eyes while the examination was in progress. Our study relied exclusively on data with the quality label "ok" and without any movement or blinking errors.

Postoperative assessment included measurements of CV and CT and was obtained at 3 and 12 months postoperatively.

## Surgery and postoperative medication

Surgical techniques were the same for all patients of one group. In the LASIK group, a hinged corneal flap with a thickness of 110 μm and a diameter of 8.4 to 8.5 mm was created with the VisuMax® femtosecond laser (Carl Zeiss Meditec, Jena, Germany). The Mel 80 excimer laser (Carl Zeiss Meditec, Jena, Germany) with iris recognition software was used for photoablation within a 6.5–6.75 treatment area. In contrast to LASIK, SMILE is a flapless procedure. Before starting the surgery, the patient was asked to fixate on a blinking target. When adequate centration was achieved, the eye was fixated using a curved suction contact glass. A lenticule was produced using the VisuMax® femtosecond laser system (Carl Zeiss Meditec, Jena, Germany) that was set at an intended cap thickness of 120 μm and optical zone of 6.25–6.5 mm. The lenticule was then grasped and removed through a small incision. All surgeries were uneventful and no severe postoperative complications occurred.

Postoperatively, patients in both groups received polymyxin/ neomycin/ dexamethasone eye-drops (Isopto-Max, Alcon) 4 times daily for 5 days. For 4 weeks, artificial tear supplements were prescribed, starting hourly the first week and reduced to weekly as needed.

## Evaluation

Assessment included changes in corneal volume (ΔCV) and corneal thickness (ΔCT). They were calculated as differences of 3-month postoperative minus preoperative measurements ($\Delta CV_{3mo}$, $\Delta CT_{3mo}$) and 12-month postoperative minus preoperative values ($\Delta CV_{12mo}$, $\Delta CT_{12mo}$). For each group, mean values and standard deviations of the evaluated outcomes were calculated. Differences of ΔCV and ΔCT between both groups were compared.

Statistical analysis was performed with Stata software. Normality of all data samples was checked with the Shapiro-Wilk test. Differences between the 2 groups were evaluated using independent sample t-tests for data following normal distribution. Where normal distribution criteria were not met, the Mann-Whitney U test was applied. Tests, for each group separately, were employed as follows: the Mann-Whitney U test was performed to compare differences between 3- and 12-month measurements; and the Pearson's correlation coefficient (r) was applied to assess the correlation between the preoperative spherical equivalent (SE) and $\Delta CV_{3mo}$ of the 7 mm diameter region, as well as the correlation between the SE and central $\Delta CT_{3mo}$. A P-value less than 0.05 was considered statistically significant.

## Results

In our study, 104 eyes of 52 patients were analyzed. Table 1 shows the characteristics of the study population. Preoperative measurements of cylinder and SE were similar in both groups as well as corneal volumes and pachymetry values. The performed refractive corrections and refractive outcomes were also equal in both groups.

Linear correlation between the SE refraction to be corrected and $\Delta CV_{3mo}$ of the 7 mm diameter zone in the LASIK group (r = 0.67, P = .000) and in the SMILE group (r = 0.31, P = .025) was found. Fig 1 shows that there is a strong relationship between the preoperative SE and apical $\Delta CT_{3mo}$ in the LASIK group (r = 0.80, P = .000) and the lenticule extraction group (r = 0.52, P = .001).

**Table 1. Study population characteristics (n = 104).**

| Parameter | Mean ± SD [Range] | | P-value |
|---|---|---|---|
| | LASIK | SMILE | |
| Number of patients/ eyes | 26 / 52 | 26 / 52 | |
| Gender: female/ male (%) | 44.23 / 55.77 | 50.00 / 50.00 | |
| Age (y) | 36.15 ± 9.29 | 34.27 ± 7.22 | |
| | [21 to 58] | [24 to 50] | |
| Eye used: right/ left (%) | 50.00 / 50.00 | 51.92 / 48.08 | |
| Sphere (D) | -3.32 ± 1.87 | -3.85 ± 1.33 | |
| | [-7-75 to 0.00] | [-6.50 to -1.00] | |
| Cylinder (D) | -0.96 ± 1.08 | -0.87 ± 0.61 | .460 |
| | [-4.50 to 0.00] | [-2.25 to 0.00] | |
| SE (D) | -3.96 ± 1.67 | -4.06 ± 1.50 | .449 |
| | [-8.00 to -1.75] | [-7.00 to -1.13] | |
| Corneal volume (mm$^3$) | | | |
| 3-mm diameter region | 4.02 ± 0.18 | 4.02 ± 0.22 | .684 |
| | [3.7 to 4.4] | [3.7 to 4.5] | |
| 5-mm diameter region | 11.76 ± 0.52 | 11.77 ± 0.62 | .897 |
| | [10.9 to 12.8] | [10.8 to 13.2] | |
| 7-mm diameter region | 25.19 ± 1.11 | 25.32 ± 1.30 | .680 |
| | [23.3 to 27.6] | [23.1 to 28.0] | |
| 10-mm diameter region | 61.36 ± 2.82 | 62.59 ± 3.43 | .114 |
| | [55.7 to 67.5] | [56.2 to 69.7] | |
| Corneal thickness (µm) | | | |
| Apex | 557.57 ± 24.54 | 556.75 ± 30.97 | .746 |
| | [516 to 604] | [512 to 631] | |
| 2-mm diameter region | 562.75 ± 24.42 | 562.52 ± 31.11 | .779 |
| | [520 to 609] | [518 to 636] | |
| 4-mm diameter region | 589.35 ± 25.26 | 590.21 ± 31.21 | .911 |
| | [548 to 641] | [542 to 662] | |
| 6-mm diameter region | 636.55 ± 27.60 | 639.33 ± 31.69 | .721 |
| | [593 to 692] | [583 to 706] | |
| 8-mm diameter region | 708.51 ± 31.38 | 717.17 ± 34.82 | .212 |
| | [653 to 773] | [645 to 783] | |

LASIK = laser-assisted in situ keratomileusis; SD = standard deviation; SE = spherical equivalent; SMILE = small-incision lenticule extraction; y = years, D = diopter.

Main outcomes and corresponding P-values are presented in Tables 2 and 3. Higher ΔCV and ΔCT were seen using SMILE. Comparing differences between both techniques, $\Delta CV_{3mo}$ of the 3, 5, and 7 mm diameter regions (P = .001 respectively) and $\Delta CV_{12mo}$ of the 5 and 7 mm diameter zones (P < .05) approached significance, as well as $\Delta CT_{3mo}$ and $\Delta CT_{12mo}$ for the central and 2, 4, and 6 mm diameter areas (P < .05).

Comparing total CV and CCT between 3 and 12 months postoperatively, we found no statistically significant difference in the LASIK group (P = .16 and .11, respectively) as well as in the SMILE group (P = .10 and .09, respectively).

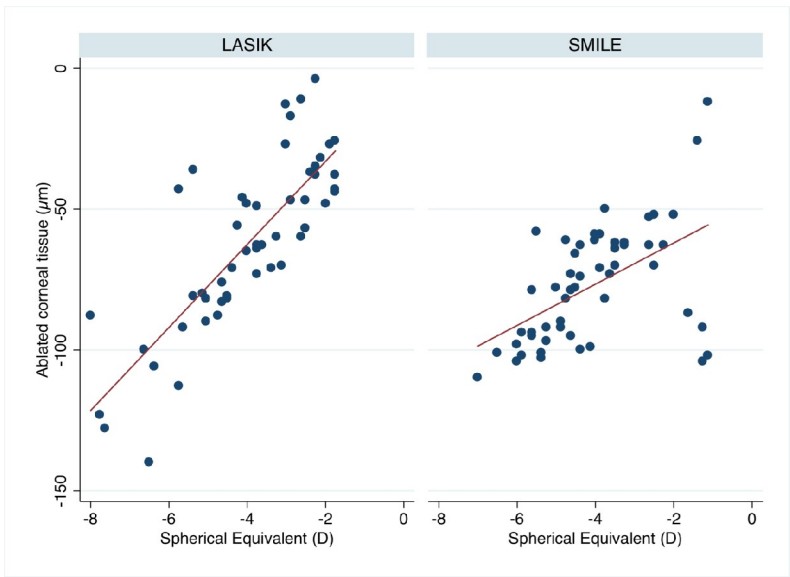

**Fig 1. Correlation between the preoperative spherical equivalent and the alteration in corneal thickness (3-month postoperative minus preoperative values).**

**Table 2. Differences of 3-month postoperative minus preoperative measurements.**

| Parameter | Mean ± SD [Range] | | P-value |
|---|---|---|---|
| | LASIK (n = 52) | SMILE (n = 52) | |
| Δ Corneal volume (mm³) | | | |
| 3-mm diameter region | -0.40 ± 0.21 | -0.50 ± 0.15 | .001 |
| | [-0.9 to 0.0] | [-0.7 to 0.0] | |
| 5-mm diameter region | -0.95 ± 0.49 | -1.21 ± 0.34 | .001 |
| | [-2.2 to 0.0] | [-1.7 to -0.1] | |
| 7-mm diameter region | -1.33 ± 0.81 | -1.77 ± 0.55 | .001 |
| | [-3.4 to 0.3] | [-2.8 to 0.1] | |
| 10-mm diameter region | -1.35 ± 1.31 | -1.78 ± 1.09 | .066 |
| | [-4.2 to 1.8] | [-4.8 to 1.5] | |
| Δ Corneal thickness (μm) | | | |
| Apex | -62.00 ± 30.54 | -77.12 ± 20.92 | .002 |
| | [-140 to -4] | [-110 to -12] | |
| 2-mm diameter region | -57.82 ± 27.95 | -70.69 ± 19.81 | .003 |
| | [-126 to -4] | [-99 to -9] | |
| 4-mm diameter region | -42.55 ± 22.26 | -54.19 ± 15.94 | .001 |
| | [-99 to 1] | [-79 to -4] | |
| 6-mm diameter region | -17.94 ± 15.06 | -25.19 ± 11.17 | .009 |
| | [-53 to 20] | [-50 to 8] | |
| 8-mm diameter region | -0.80 ± 12.80 | -3.13 ± 12.05 | .418 |
| | [-7 to 30] | [-31 to 28] | |

LASIK = laser-assisted in situ keratomileusis; SD = standard deviation; SMILE = small-incision lenticule extraction; Δ = 3-month postoperative minus preoperative.

**Table 3. Differences of 12-month postoperative minus preoperative measurements.**

| Parameter | Mean ± SD [Range] | | P-value |
|---|---|---|---|
| | LASIK (n = 52) | SMILE (n = 52) | |
| Δ Corneal volume (mm$^3$) | | | |
| 3-mm diameter region | -0.42 ± 0.22 | 0.49 ± 0.14 | .053 |
| | [-1.0 to 0.0] | [-0.7 to -0.1] | |
| 5-mm diameter region | -0.99 ± 0.52 | -1.17 ± 0.34 | .023 |
| | [-2.5 to 0.1] | [-1.7 to -0.2] | |
| 7-mm diameter region | -1.41 ± 0.85 | -1.69 ± 0.55 | .020 |
| | [-4.0 to 0.7] | [-2.9 to -0.2] | |
| 10-mm diameter region | -1.39 ± 1.32 | -1.62 ± 1.10 | .333 |
| | [-5.2 to 2.7] | [-5.2 to 0.7] | |
| Δ Corneal thickness (μm) | | | |
| Apex | -64.00 ± 32.65 | -75.62 ± 20.33 | .013 |
| | [-154 to -5] | [-108 to -18] | |
| 2-mm diameter region | -59.59 ± 30.15 | -69.37 ± 18.96 | .024 |
| | [-140 to -3] | [-99 to -15] | |
| 4-mm diameter region | -43.98 ± 24.12 | -52.60 ± 15.58 | .024 |
| | [-112 to 10] | [-81 to -8] | |
| 6-mm diameter region | -18.61 ± 15.21 | -23.79 ± 11.48 | .022 |
| | [-66 to 28] | [-54 to 4] | |
| 8-mm diameter region | -0.65 ± 11.82 | -2.33 ± 12.43 | .639 |
| | [-27 to 38] | [-34 to 20] | |

LASIK = laser-assisted in situ keratomileusis; SD = standard deviation; SMILE = small-incision lenticule extraction; Δ = 12-month postoperative minus preoperative.

## Discussion

### Changes of corneal volume and thickness

Significant differences in the statistics between both groups were found at $\Delta CV_{3mo}$ and $\Delta CT_{3mo}$. Alterations in partial CV of regions with 3, 5, and 7 mm diameters were significantly larger in the lenticule extraction group (P = .001). Changes in CT were comparatively higher in the SMILE group than in the LASIK group. This was also found to be significant for the apical pachymetry and the 2, 4, and 6 mm diameter zones (P < .01). Comparing total $\Delta CV_{3mo}$ and $\Delta CT_{3mo}$ in the 8 mm diameter area, no significant difference was found. Alterations remained stable after 12 months postoperatively.

One possibility that explains why the SMILE technique achieves greater alterations in CV and CT is the fact that the maximal depth in LASIK is ablated in the central treatment zone [11]. Due to the laser beam being oval-shaped, less tissue ablation occurs towards the periphery [12]. Another possibility may be different wound healing processes. Between the flap stroma and the ablated area keratocytes produce a lamellar corneal stromal scar [13]. The keratocyte-mediated stromal production can be stated to increase towards the periphery as this process at the margin is necessary to fixate the flap [13]. Therefore, a strong, hypercellular and fibrotic corneal stromal scar is clinically seen in the periphery at flap wound margins, whereas the primitive hypocellular scar in the center is not visible postoperatively [13, 14]. Since the corneal stroma is created postoperatively, it seems like the intraoperative ablation volume is smaller. In contrast, Luft et al. [10] found comparable levels of keratocyte apoptosis and proliferation in the ex vivo human corneas after LASIK and SMILE. Similar findings were reported by Riau et al. in a rabbit model [15].

Using the SMILE procedure, apical $\Delta CT_{3mo}$ was higher than in the LASIK group (64.5 μm versus 38.3 μm) in the study by Kobashi et al. [7] which was congruent with our results. Another study [16] reported central $\Delta CT_{3mo}$ to be larger using LASIK (89.3 μm) compared to SMILE (81.2 μm).

Furthermore, especially in correction of very low refractive errors there is a highly variability in loss of corneal thickness as seen in Fig 1. We assume that this is due to the high level of mechanical manipulation during SMILE which makes more likely that higher variations occur. Future studies need to evaluate how good the flapless technique performs due to this specifically for the correction of very low refractive errors around -1 dpt.

## Impact on corneal biomechanical properties

The cornea is an anisotropic tissue with interlamellar branching collagen fibers that is stronger in the periphery than in the center [17]. Moreover, the vital collagen bundles in the anterior stroma are stiffer than in the posterior part and the interconnectivity of the lamellae in the anterior stroma provides the main biomechanical strength to the cornea [17, 18]. These strong anterior lamellae are preserved and remain intact during flapless surgery. In contrast, flap-based LASIK effects the corneal biomechanics more than flapless SMILE as it impairs the strong anterior stroma overlying the exposed stromal bed [19]. Besides lifting the flap before excimer laser ablation, the stroma is exposed to hydration changes that fracture the stability needed for exact refractive correction [7]. Overall, only considering the surgical procedure, corneal biomechanical strength is better preserved after SMILE as it causes less disruption of the corneal structural integrity [13].

Corneal hysteresis (CH) and corneal resistance factor (CRF), which can be obtained from the Ocular Response Analyzer, illustrate viscoelastic properties of the cornea and thus are a measure of stability. We did not analyze CH and CRF in our patients. However, we can draw assumptions about the biomechanical properties based on the changes of CT and CV. The more volume that is ablated with LASIK, the more CH and CRF are reduced [20]. Lower values stand for a biomechanically weaker cornea.

At short notice, LASIK is likely to show reduced CH and CRF due to wound healing processes, whereas SMILE is supposed to have larger CH and CRF as the cornea is unimpaired. We assessed our measurements at 3 months postoperatively. Hence, we expect the wound healing processes not to have any influence on our results.

In general, a higher CCT results in a stiffer appearance of the cornea [21]. We found a lower CCT after SMILE, significantly more reduced than after LASIK. Logically, we should expect greater decrement of CH and CRF in the lenticule extraction group which means that biomechanical properties have been more affected by the flapless technique. One possible explanation for this finding is that the removed lenticule volume was larger than the ablation volume with LASIK. Although less stromal collagen fibers are severed in SMILE, it does not mean that corneal rigidity is unconditionally higher after flapless surgery. It must be pointed out that, after having removed the lenticule, the length of the back of the anterior cap will be longer than the corresponding arc length of the unaffected residual stroma. Hence, differences in arc lengths contribute to a corrugated contact surface. Since a rearrangement of anterior collagen lamellae is necessary, the unimpaired cap is not as strong as expected [21]. The more volume that is removed, the greater the differences in arc lengths, and hence the corrugation. The more decreased CCT after SMILE than after LASIK causes the necessity of augmented rearrangement that may reduce corneal stability. To date, several studies have evaluated changes in CH and CRF after LASIK and SMILE. Most studies report similar reductions of CH and CRF values in both groups [19, 21, 22]. Wang et al. [19] found a greater reduction of CRF with flap-based

surgery, but differences between the 2 groups were not statistically significant. It should be remarked that collagen crosslinking modifies corneal strength significantly, whereas CH and CRF might be unaffected [23]. Therefore, an absence of differences in CH and CRF between both techniques is not equivalent to an absence of biomechanical alteration. Corneal properties might be different after the two procedures even if CH and CRF remain similar.

With regard to our previous considerations, we cannot state definitively whether corneal structural integrity is more affected by LASIK or SMILE.

## Long-term examination of total CV and CCT

Comparing total CV and CCT measured at 3 and 12 months postoperatively, we found no significant difference in each group which illustrates that alterations in volume and pachymetry occurred during the first 3 months. We suppose that most of the measured $\Delta CV_{3mo}$ correspond with intraoperative ablation volume and lenticule volume, respectively. Except for wound healing procedures, we did not expect the volume to change much postoperatively. Later on, the CV and CCT stayed approximately constant and we do not expect them to alter further. This was similar to the findings by Kobashi et al. [7] that stated CCT remain unchanged during the postoperative period.

## Limitations

Our study has several limitations. The main deficiency is that the study was not a prospective and randomized controlled study. Results might have been different if we had matched the patients preoperatively and referred them to a certain group by randomization.

One aspect that may have influenced the results is the difference between LASIK flap thickness (110 μm) and SMILE cap thickness (120 μm). It is not clear whether extracting corneal material from superficial or deeper stroma obtains better results. On one hand and as mentioned before, the deeper the intervention, the less influence on anterior corneal properties. Generally, with SMILE, the excess tissue is removed from deeper corneal stroma. Since the stronger anterior stroma stays untouched, it seems reasonable to assume greater biomechanical strength after SMILE than after LASIK. On the other hand, the deeper the lenticule is removed more volume has to be extracted to result in similar changes in the shape of the anterior corneal surface. That means, removing a lenticule from deeper stroma, more material has to be extracted to achieve the same refractive correction as with a superficial lenticule [22]. For LASIK, increased flap thickness leads to decreased corneal stability owing to intensified wound healing processes at the margin and lower residual stromal bed thickness [24]. In our study, cap and flap thicknesses were intentionally not equally set in order to maintain standard parameters usually used in the clinical setting.

We measured intraocular pressure before and after intervention, but as there were no pressure spikes it was not noted in the patients' files. Therefore, we were not able to analyse changes in intraocular pressure (IOP) before and after intervention in our study, even though this is a widely discussed issue after refractive surgery. SMILE is reported to be less sensitive to IOP changes due to higher biomechanical stability [14]. We could not take a position on this issue.

## Conclusions

Although as previous reports have shown the flapless technique has similar effect on corneal stability in the short term we advise not to go beyond current limits of recommended ranges for correction of refractive errors, as loss with CCT and corneal volume is higher with SMILE. Long term studies are required to evaluate the potential higher incidence of corneal ectasia in the flapless technique due to this.

## Author Contributions

**Conceptualization:** Anna Schuh, Wolfgang J. Mayer, Efstathios Vounotrypidis, Thomas Kreutzer, Thomas Kohnen, Siegfried Priglinger, Mehdi Shajari, Daniel Kook.

**Data curation:** Anna Schuh, Carolin M. Kolb, Mehdi Shajari, Daniel Kook.

**Formal analysis:** Anna Schuh, Carolin M. Kolb, Daniel Kook.

**Investigation:** Anna Schuh, Wolfgang J. Mayer, Efstathios Vounotrypidis, Mehdi Shajari, Daniel Kook.

**Methodology:** Anna Schuh.

**Project administration:** Anna Schuh, Daniel Kook.

**Supervision:** Wolfgang J. Mayer, Mehdi Shajari, Daniel Kook.

**Validation:** Carolin M. Kolb, Wolfgang J. Mayer, Efstathios Vounotrypidis, Thomas Kreutzer, Thomas Kohnen, Siegfried Priglinger, Mehdi Shajari, Daniel Kook.

**Visualization:** Carolin M. Kolb, Daniel Kook.

**Writing – original draft:** Anna Schuh.

**Writing – review & editing:** Anna Schuh, Wolfgang J. Mayer, Efstathios Vounotrypidis, Thomas Kreutzer, Thomas Kohnen, Siegfried Priglinger, Mehdi Shajari, Daniel Kook.

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
