## [Decision Letter · Decision Letter 0]

4 Nov 2020

PONE-D-20-24719

Comparison of changes in corneal volume and corneal thickness after myopia correction between LASIK and SMILE

PLOS ONE

Dear Dr. %Dr.  Shajari%,

Thank you for submitting your manuscript to PLOS ONE. After careful consideration, we feel that it has merit but does not fully meet PLOS ONE’s publication criteria as it currently stands. Therefore, we invite you to submit a revised version of the manuscript that addresses the points raised during the review process.

The reviewers raised minor concerns with your MSS, and I do agree with those comments. Your appropriate addressing of them will allow EBM to render decision on your MSS.  

We look forward to receiving your revised manuscript.

Kind regards,

Rajiv R. Mohan, Ph.D.

Academic Editor

PLOS ONE

Journal Requirements:

'The authors received no specific funding for this work.'

We note that one or more of the authors are employed by a commercial company: SMILE Eyes Eye Clinic

Reviewers' comments:

Reviewer's Responses to Questions

**Comments to the Author**

1. Is the manuscript technically sound, and do the data support the conclusions?

Reviewer #1: Yes

Reviewer #2: Yes

2. Has the statistical analysis been performed appropriately and rigorously? 

Reviewer #1: Yes

Reviewer #2: Yes

3. Have the authors made all data underlying the findings in their manuscript fully available?

Reviewer #1: Yes

Reviewer #2: Yes

4. Is the manuscript presented in an intelligible fashion and written in standard English?

Reviewer #1: Yes

Reviewer #2: Yes

5. Review Comments to the Author

Reviewer #1: In assessing residual thickness and its overall effects you have omitted data relating to intraocular pressure obtained prior to intervention as well as subsequent evaluations. This data should be available and if so included in the presentation with appropriate discussion as to its effects. If not available the discussion should indicate its absence as well as potential positive and negative effects.

Reviewer #2: Comparison of changes in corneal volume and corneal thickness after myopia correction between LASIK and SMILE.

Schuh et al. reported the changes in corneal volume and corneal thickness after myopia correction between LASIK and SMILE.

The authors reported that loss in central corneal thickness and corneal volume with SMILE is higher than LASIK.

The manuscript requires the following minor corrections.

Introduction: Please include histological and ultrastructural studies of the cornea treated with SMILE and LASIK.

Discussion

The discussion is very long. Please discuss the differences in CV, CT, CCT, CH, and CRF between SMIL and LASIK step by step.

Line 221: ‘…less tissue ablation occurs towards the periphery'. Please write a reference if it is not the author's results.

Line 223: ‘keratocytes produce a lamellar corneal stromal scar'. The statement requires histological assessment. If the authors did not do any histological assessment, then please quote reference. Luft et al. 2018 reported that the keratocyte proliferation and apoptosis in the human donor eye model were similar in SMILE and LASIK. Please discuss relating to your work.

Line 228: ‘…a strong ..postoperatively’. Is there any histological reference regarding the statement?

Line 253-255: The statement requires a reference. Please discuss the interconnectivity of the lamellae in the anterior stroma in relation to the LASIk.

6. PLOS authors have the option to publish the peer review history of their article (what does this mean?). If published, this will include your full peer review and any attached files.

Reviewer #1: No

Reviewer #2: No

---

## [Author Response · Author response to Decision Letter 0]

19 Mar 2021

Dear Professor Mohan,

Dear Editorial board,

Thank you for the opportunity to improve our paper. Please find our responses to the journal requirements and reviewers below.

I. Journal Requirements:

Style requirements were changed accordingly. 

2. We note that you have indicated that data from this study are available upon request. PLOS only allows data to be available upon request if there are legal or ethical restrictions on sharing data publicly. 

We uploaded the data on figshare. 

DOI: https://doi.org/10.6084/m9.figshare.14186699.v1

3. Thank you for stating the following in the Financial Disclosure section: 'The authors received no specific funding for this work.' We note that one or more of the authors are employed by a commercial company: SMILE Eyes Eye Clinic

a. Funding Statement:

The funder provided support in the form of salaries for authors [DK], but did not have any additional role in the study design, data collection and analysis, decision to publish, or preparation of the manuscript. The specific roles of this author are articulated in the ‘author contributions’ section.

b. Competing Interests Statement:

There are no financial or non-financial competing interests to declare of any of the authors. One of the authors [DK] is employed by the commercial company SMILE Eyes Eye Clinic. However, this brought no competing interests, no financial or non-financial, professional or personal, and does not alter our adherence to PLOS ONE policies on sharing data and materials. 

See 3.a. and 3.b.

ORCID iD Mehdi Shajari: 0000-0003-3961-4168.

II. Reviewers' comments:

1. – 4. No comments.

5. Review Comments to the Author

Reviewer #1: 

In assessing residual thickness and its overall affects you have omitted data relating to intraocular pressure obtained prior to intervention as well as subsequent evaluations. This data should be available and if so, included in the presentation with appropriate discussion as to its effects. If not available, the discussion should indicate its absence as well as potential positive and negative effects.

We thank the reviewer for reviewing our manuscript and for the kind feedback. It is correct that changes in intraocular pressure (IOP) after refractive treatment of the cornea is a known and widely discussed issue, especially regarding the question whether SMILE is superior to LASIK. We did check patients’ IOP before and after the intervention. However, it was only noted if there was increased IOP postoperatively. Therefore, we were not able to analyse IOP changes. It is correct, that this should be mentioned in the discussion part, so we changed it accordingly (p. 18):

We measured intraocular pressure before and after intervention, but as there were no pressure spikes it was not noted in the patients’ files. Therefore, we were not able to analyse changes in intraocular pressure (IOP) before and after intervention in our study, even though this is a widely discussed issue after refractive surgery. SMILE is reported to be less sensitive to IOP changes due to higher biomechanical stability.[14] We could not take a position on this issue.

Reviewer #2: 

Comparison of changes in corneal volume and corneal thickness after myopia correction between LASIK and SMILE.

Schuh et al. reported the changes in corneal volume and corneal thickness after myopia correction between LASIK and SMILE.

The authors reported that loss in central corneal thickness and corneal volume with SMILE is higher than LASIK.

The manuscript requires the following minor corrections.

 Thank you, a lot, for your helpful and constructive comments.

Introduction: Please include histological and ultrastructural studies of the cornea treated with SMILE and LASIK.

 This was added to the introduction (p. 4):

Regarding ultrastructural changes of the cornea after refractive interventions, Luft et al.[10] observed that reactive fibrosis was less marked after SMILE compared to LASIK in a human donor eye model, however after LASIK the stromal bed exhibited a smoother surface texture.

Discussion

The discussion is very long. Please discuss the differences in CV, CT, CCT, CH, and CRF between SMIL and LASIK step by step.

 We shortened the discussion. Please find the changes on p. 12-16.

Line 221: ‘…less tissue ablation occurs towards the periphery'. Please write a reference if it is not the author's results.

 A reference to this statement has been added (p.13): 

Gatinel D, Hoang-Xuan T, Azar DT. Volume estimation of excimer laser tissue ablation for correction of spherical myopia and hyperopia. Invest Ophthalmol Vis Sci. 2002;43(5):1445-9. Epub 2002/05/01. PubMed PMID: 11980859.

Line 223: ‘keratocytes produce a lamellar corneal stromal scar'. The statement requires histological assessment. If the authors did not do any histological assessment, then please quote reference. Luft et al. 2018 reported that the keratocyte proliferation and apoptosis in the human donor eye model were similar in SMILE and LASIK. Please discuss relating to your work.

1. A reference was added (p.13):

Between the flap stroma and the ablated area keratocytes produce a lamellar corneal stromal scar.[13]

Dawson DG, Kramer TR, Grossniklaus HE, Waring GO, 3rd, Edelhauser HF. Histologic, ultrastructural, and immunofluorescent evaluation of human laser-assisted in situ keratomileusis corneal wounds. Arch Ophthalmol. 2005;123(6):741-56. Epub 2005/06/16. doi: 10.1001/archopht.123.6.741. PubMed PMID: 15955975.

2. We disscussed the study of Luft et al. as well as Riau et al. accordingly (p.13):

Another possibility may be different wound healing processes. Between the flap stroma and the ablated area keratocytes produce a lamellar corneal stromal scar.[13] The keratocyte-mediated stromal production can be stated to increase towards the periphery as this process at the margin is necessary to fixate the flap.[13] Therefore, a strong, hypercellular and fibrotic corneal stromal scar is clinically seen in the periphery at flap wound margins, whereas the primitive hypocellular scar in the center is not visible postoperatively.[13, 14] Since the corneal stroma is created postoperatively, it seems like the intraoperative ablation volume is smaller. In contrast, Luft et al.[10] found comparable levels of keratocyte apoptosis and proliferation in the ex vivo human corneas after LASIK and SMILE. Similar findings were reported by Riau et al. in a rabbit model.[15]

Line 228: ‘…a strong ..postoperatively’. Is there any histological reference regarding the statement?

We refer to the histological study of Dawson et al. for this statement, which was also given as a reference. The reference at the end of the sentence refers to the whole statement.

Dawson DG, Kramer TR, Grossniklaus HE, Waring GO, 3rd, Edelhauser HF. Histologic, ultrastructural, and immunofluorescent evaluation of human laser-assisted in situ keratomileusis corneal wounds. Arch Ophthalmol. 2005;123(6):741-56. Epub 2005/06/16. doi: 10.1001/archopht.123.6.741. PubMed PMID: 15955975.

Line 253-255: The statement requires a reference. Please discuss the interconnectivity of the lamellae in the anterior stroma in relation to the LASIk.

 We added two references

Winkler M, Shoa G, Xie Y, Petsche SJ, Pinsky PM, Juhasz T, et al. Three-dimensional distribution of transverse collagen fibers in the anterior human corneal stroma. Invest Ophthalmol Vis Sci. 2013;54(12):7293-301. Epub 2013/10/12. doi: 10.1167/iovs.13-13150. PubMed PMID: 24114547; PubMed Central PMCID: PMCPMC4589141.

Petsche SJ, Chernyak D, Martiz J, Levenston ME, Pinsky PM. Depth-dependent transverse shear properties of the human corneal stroma. Invest Ophthalmol Vis Sci. 2012;53(2):873-80. Epub 2011/12/30. doi: 10.1167/iovs.11-8611. PubMed PMID: 22205608; PubMed Central PMCID: PMCPMC3317426.

 and changed the discussion accordingly (p.14):

The cornea is an anisotropic tissue with interlamellar branching collagen fibers that is stronger in the periphery than in the center.[17] Moreover, the vital collagen bundles in the anterior stroma are stiffer than in the posterior part and the interconnectivity of the lamellae in the anterior stroma provides the main biomechanical strength to the cornea.[17, 18] These strong anterior lamellae are preserved and remain intact during flapless surgery. In contrast, flap-based LASIK effects the corneal biomechanics more than flapless SMILE as it impairs the strong anterior stroma overlying the exposed stromal bed.[19]

Sincerely,

Mehdi Shajari

---

## [Editor Report · Decision Letter 1]

13 Apr 2021

Comparison of changes in corneal volume and corneal thickness after myopia correction between LASIK and SMILE

PONE-D-20-24719R1

Dear Dr. %Shajari%,

We’re pleased to inform you that your manuscript has been judged scientifically suitable for publication and will be formally accepted for publication once it meets all outstanding technical requirements.

Kind regards,

Rajiv R. Mohan, Ph.D.

Academic Editor

PLOS ONE
---

## [Editor Report · Acceptance letter]

23 Apr 2021

PONE-D-20-24719R1 

Comparison of changes in corneal volume and corneal thickness after myopia correction between LASIK and SMILE 

Dear Dr. Shajari:

I'm pleased to inform you that your manuscript has been deemed suitable for publication in PLOS ONE. Congratulations! Your manuscript is now with our production department. 

Kind regards, 

on behalf of

Dr. Rajiv R. Mohan 

Academic Editor

PLOS ONE